# Light chain mutations contribute to defining the fibril morphology in systemic AL amyloidosis

Sara Karimi-Farsijani[1,4], Peter Benedikt Pfeiffer [1,4] ✉, Sambhasan Banerjee [1], Julian Baur[1], Lukas Kuhn[1], Niklas Kupfer[1], Ute Hegenbart [2], Stefan O. Schönland [2], Sebastian Wiese [3], Christian Haupt[1], Matthias Schmidt [1] & Marcus Fändrich [1]

Systemic AL amyloidosis is one of the most frequently diagnosed forms of systemic amyloidosis. It arises from mutational changes in immunoglobulin light chains. To explore whether these mutations may affect the structure of the formed fibrils, we determine and compare the fibril structures from several patients with cardiac AL amyloidosis. All patients are affected by light chains that contain an *IGLV3-19* gene segment, and the deposited fibrils differ by the mutations within this common germ line background. Using cryo-electron microscopy, we here find different fibril structures in each patient. These data establish that the mutations of amyloidogenic light chains contribute to defining the fibril architecture and hence the structure of the pathogenic agent.

Systemic AL amyloidosis is one of the most frequently diagnosed forms of systemic amyloidosis[1]. The disease arises from the misfolding and aggregation of monoclonal antibody light chains (LCs)[2]. LCs are proteins from the adaptive immune system and are naturally hypervariable, which allows them to bind to a diverse range of antigens. The hypervariability arises from the combination of three types of gene segments, termed variable (V), joining (J) and constant (C), that are encoded in the germ line (GL), and the further modification of the recombined LC genes by somatic hypermutation[3,4]. This process inserts mutations at different positions of the sequence, but mainly in the complementary determining regions (CDRs), as the antibody undergoes affinity maturation. Due to the hypervariability of antibody LCs, systemic AL amyloidosis is a patient-specific disease. Almost every patient is affected by a sequentially different, monoclonal LC[5], and there are vast differences in the exact clinical and pathological disease manifestations, such as in the organ involvement and severity of the disease[1].

Several studies previously established that the LC sequence is crucial for determining the disease onset and manifestations. λ-LCs are more abundantly found in AL amyloidosis than in normal antibodies, which mainly involve κ-LCs[6]. The *IGLV1-44* gene segment is preferentially associated with heart involvement[7], while the *IGLV6-57* segment is predominantly seen in patients with renal AL amyloid deposits[8]. The mutations of amyloidogenic LCs (that is, the residues that differ in the patient proteins from the GL) are typically destabilising to the native fold of antibody LCs[3,9] and promote the formation of fibrils by increasing the fraction of relatively unfolded but highly aggregation-prone protein states[10–13]. Whether the mutations additionally contribute to the disease by inducing a specific fibril state, is so far unknown.

Available cryo-EM structures of AL amyloid fibrils do not suffice to resolve this issue. They were obtained from four different AL patients (FOR001, FOR005, FOR006 and AL-55) with a similar clinical phenotype that involved severe cardiomyopathy. The fibrils exhibited significant differences in structure[13–16] as well as in the amino acid sequence of the LCs giving rise to these fibrils. These differences encompassed not only different mutations but also different V gene segments (*IGLV1-51*, *IGLV3-19*, *IGLV1-44* and *IGLV6-57*). Therefore, it

[1]Institute of Protein Biochemistry, Ulm University, Ulm, Germany. [2]Medicinal Department V, Amyloidosis Centre, Heidelberg University Hospital, Heidelberg, Germany. [3]Core Unit Mass Spectrometry and Proteomics, Ulm University, Ulm, Germany. [4]These authors contributed equally: Sara Karimi-Farsijani, Peter Benedikt Pfeiffer. ✉e-mail: peter.pfeiffer@uni-ulm.de

remained unclear whether the different fibril structures arise solely from different GL segments or whether there is an additional contribution of the mutations.

To dissect the effects of GL segments and mutations, we here compare the fibrils from three patients with the same V segment (*IGLV3-19*). That is, the underlying fibril proteins differed in the V segment, solely by their mutational changes. Two of the structures employed in this comparison were previously obtained from patient FOR005[15]. These structures are now compared with the two obtained fibril structures that we isolated from patients FOR103 and FOR010. Our data demonstrate substantial effects of the mutations on the fibril protein fold and overall fibril topology. Hence, they play a crucial role in defining the structure of the pathogenic agent.

## Results

### Purification and primary structure determination of the FOR010 fibril protein

Amyloid fibrils were isolated from the amyloid-laden tissue of two patients with systemic AL amyloidosis. The first patient (FOR103) showed strong cardiac involvement and fibrils were isolated from a biopsy that was taken from the abdominal fat tissue. The second patient (FOR010) also showed major cardiac involvement and fibrils were purified from the cardiac tissue which became available to us due to heart transplantation. Denaturing protein gel electrophoresis showed that the two fibril samples were relatively pure and contained a dominant protein band in the fibril-containing fractions (SI Fig. 1). The amino acid sequence of the FOR103 fibril protein was obtained previously based on cDNA sequencing and mass spectrometry (MS). The precursor LC in this patient is derived from *IGLV3-19*, *IGLJ2* and *IGLC2* GL segments, similar to the LC precursor of the FOR005 fibril protein[15,17].

In the case of the FOR010 fibril protein, it was not possible to determine the sequence of the precursor LC by cDNA sequencing and

its amino acid sequence as well as its posttranslational modifications (PTMs) were determined here at the protein level using MS. To that end, the purified fibril protein was digested with different proteases followed by an MS-based detection of the proteolytic fragments. The identified fragments were then used to computationally reconstruct the amino acid sequence of the FOR010 fibril protein (SI Fig. 2). In a second step, we subjected the undigested fibril protein to MS and determined the total masses of the fibril proteins (SI Fig. 3). Compared with the amino acid sequence, the mass allowed us to identify the LC fragments that are present in the fibril proteins, along with their PTMs.

Three major protein species were identified with this approach that exhibited molecular masses of 11,587.8, 11,718.9 and 12,045.0 Da (SI Fig. 3). The three species corresponded to three LC fragments Pro8 - Ser116, Leu4 - Lys112 and Leu4 - Ser116 (SI Table. 1) and covered sequence segments from the variable LC ($V_L$) domain that is encoded by the LC V and J segments, along with few residues from the constant LC domain (SI Fig. 3). Each fibril protein is posttranslationally modified by an intramolecular disulphide bond (SI Table 1) that occurs at the canonical position of the $V_L$ domain disulphide bond (Fig. 1). Other PTMs were not observed in the three major fibril protein species, demonstrating that they are not very abundant in the fibril (SI Table 1).

### GL assignment and identification of the mutant sites

A comparison of the FOR010 fibril protein sequence with known LC V segments indicates that it is derived from an *IGLV3-19* gene segment. The same V segment underlies also the AL fibril proteins from patients FOR103 and FOR005[17,18]. In the case of the J segment, it was not possible to unambiguously identify the GL precursor without DNA-based sequence information, as the protein sequences are highly similar, if not identical, in the *IGLJ2* and *IGLJ3* GL segments. Therefore, we hereafter compare the FOR010 fibril protein sequence with *IGLJ2* GL segments that are also found in the FOR103 and FOR005 fibril proteins[17,18]. We further assumed the C region to be based on an *IGLC2* segment,

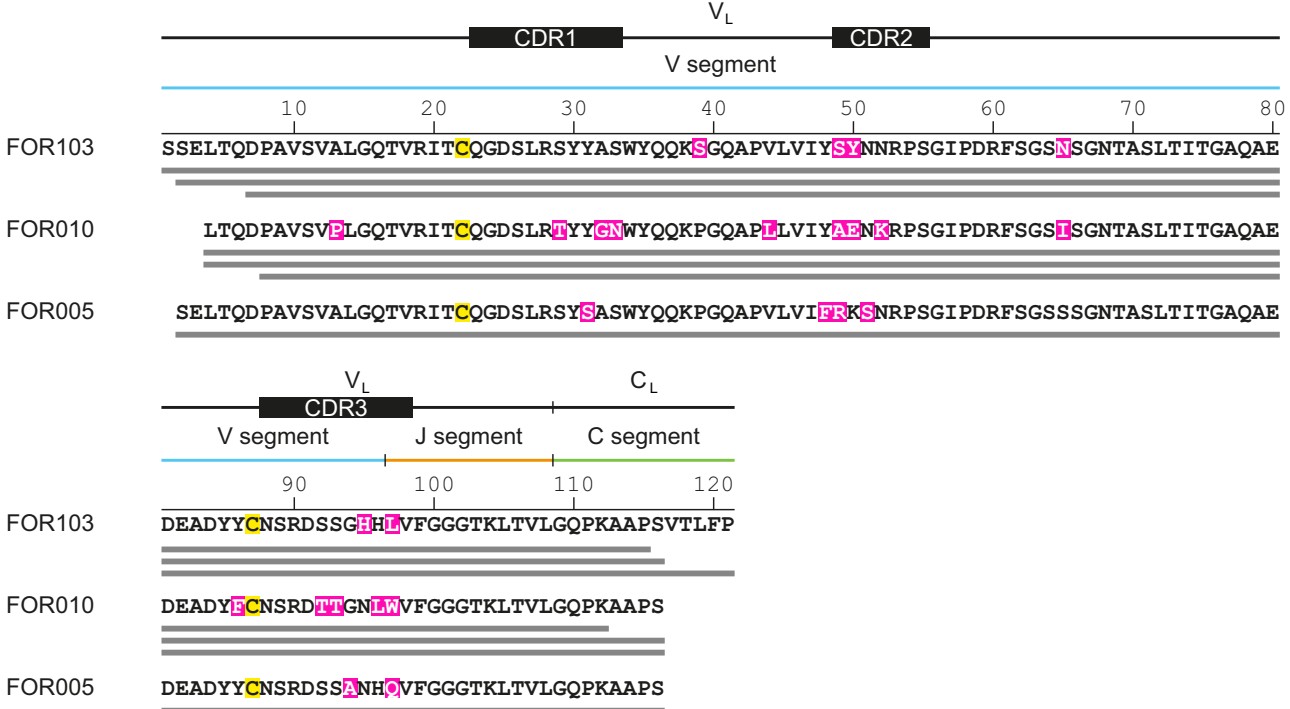

**Fig. 1 | Primary structures of the analysed fibril proteins.** Alignment of the amino acid sequences of the FOR103, FOR010 and FOR005 fibril proteins. The bars below the sequence refer to the experimentally observed fibril proteins. The FOR010 data were taken from SI Table 1. The FOR103 and FOR005 data were taken from previous studies[15,18]. Magenta: amino acid changes compared to GL sequences of *IGLV3-19* and *IGLJ2* segments; yellow: cysteine residues involved in disulphide bond formation.

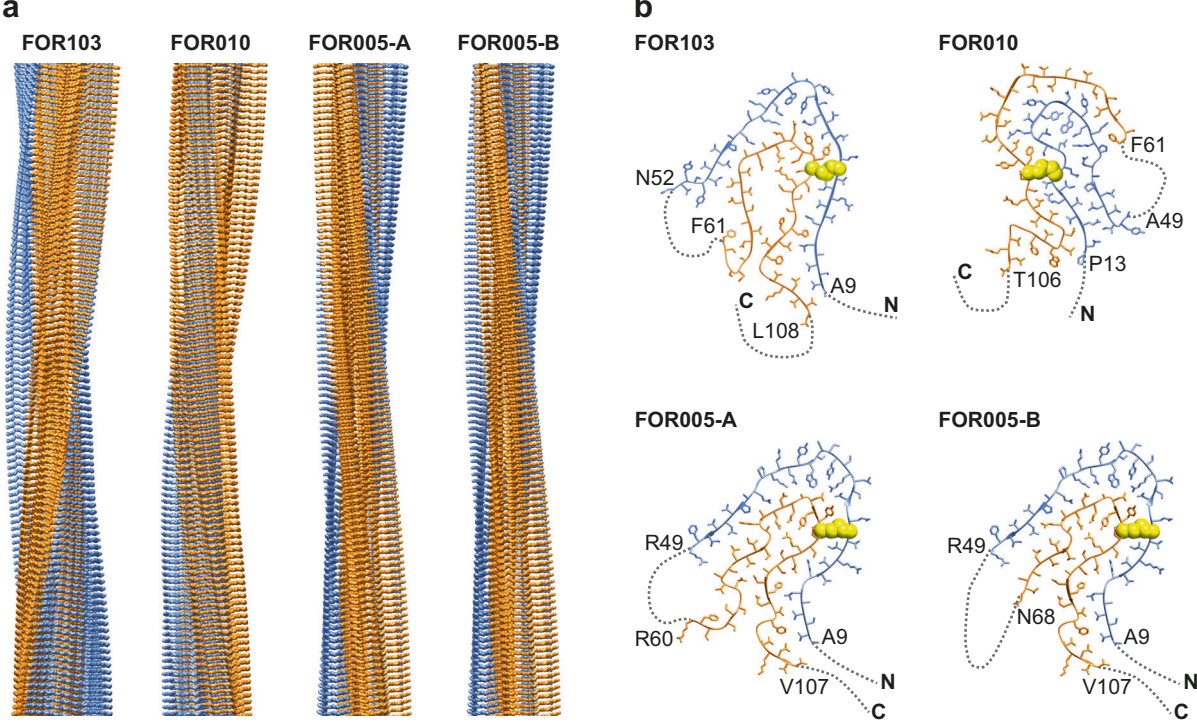

**a**

FOR103  FOR010  FOR005-A  FOR005-B

**b**

FOR103  FOR010

FOR005-A  FOR005-B

**Fig. 2 | Cryo-EM structures of *IGLV3-19*-derived AL amyloid fibrils. a** Side views of the molecular models of FOR103, FOR010 and FOR005 (A and B) fibrils. The FOR103 structure is right-hand twisted. The other fibrils possess a left-hand twist. **b** Cross-sectional views of one molecular layer of the four fibrils. The first and last residues of the ordered regions (ribbon diagrams with side chains) are highlighted.

Structurally disordered regions are represented as dotted lines. The disulphide bond is shown in a space-filled representation (yellow). The N-terminal ordered region is coloured in blue in all panels, while the C-terminal region is consistently coloured orange.

consistent with the FOR103 and FOR005 fibril proteins[17,18]. Comparison of the mutant patients' sequences with the non-mutated GL segments led to the identification of six amino acid changes in the fibril proteins of patients FOR005 and FOR103 (Fig. 1), and fourteen in the fibril protein from patient FOR010 (Fig. 1).

## Cryo-EM reconstruction and handedness of FOR103 and FOR010 fibrils

Analysis of the isolated fibrils with platinum side shadowing and scanning electron microscopy (SEM) shows that FOR010 amyloid fibrils are left-hand twisted (SI Fig. 4b), similar to FOR005 fibrils[15]. FOR103 amyloid fibrils, by contrast, are right-hand twisted (SI Fig. 4a). Cryo-EM further shows the two fibril samples contain a major fibril morphology that accounts for approximately 85 % of the fibrils in each sample (SI Fig. 5). The dominant fibril morphology from each sample was structurally reconstructed to a spatial resolution of 2.92 Å (FOR103) and 2.25 Å (FOR010), based on the 0.143 Fourier-shell correlation (FSC) criterion (SI Fig. 6, SI Table 2). The reconstructed three-dimensional (3D) maps were fitted with molecular models (SI Fig. 7) that depict polar and C1-helical fibril structures that consist of single stacks of fibril proteins or protofilaments (Fig. 2a), similar to FOR005 fibrils[15]. The model resolution was 3.1 Å in the case of the FOR103 fibril and 2.0 Å for the FOR010 fibril (SI Fig. 8, SI Table 3).

## Fibril protein fold of *IGLV3-19*-derived fibrils

All fibrils contain a stable core that is formed by two segments of the LC sequence. These segments extend roughly from residue 10 to residue 50 and from position 60 to position 110 and harbour the fibril cross-β structure (Fig. 2b). There are seven to fourteen parallel β-sheets (Fig. 3a) that are N- and C-terminally flanked by conformationally disordered regions (Fig. 2b). The disordered regions are not well resolved in the 3D maps, similar to an internal

disordered region that extends roughly from residue 50 to position 60. All fibril proteins show an amyloid key fold (Fig. 3b). The amyloid key fold resembles the Greek key motif of globular protein structures. However, the amyloid key does not form intramolecular β-sheets but is instead stabilised by intermolecular hydrogen bonds (β-sheets) along the fibril z-axis and side chain-side chain interactions of the β-strands[19,20]. This motif occurs in many amyloid fibril structures[19-21], although the outer arc of the key is structurally disordered in our fibrils and not seen in the 3D maps. In the present structures, we noted that the fibrils differ in the handedness of the swirl of the amyloid key (Fig. 3b, grey rotation arrows). The swirl is defined by the position of the central arc of the amyloid key relative to the N- and C-terminal ends of the polypeptide chain. The swirl of the key is right-handed in the FOR005 and FOR103 fibril proteins when oriented as shown in Fig. 3b, and left-handed in the FOR010 fibril protein. The swirl does not relate to the handedness of the fibril supertwist, which is left-hand for FOR005 and FOR010 fibrils and right-hand for FOR103 (Fig. 2a).

All three fibril proteins enclose a number of buried cavities, but only the FOR103 fibril protein contains a cavity that is lined by polar and ionic amino acid side chains (SI Fig. 9) and which is large enough to fit a number of water molecules. The FOR103 fibril contains even an additional cavity that is surrounded by hydrophobic amino acid residues. This cavity is filled with density that does not originate from the polypeptide chain (SI Fig. 9), indicating that it contains a hydrophobic inclusion as an integral part of the fibril structure, similar to the previously described λ1-derived AL amyloid fibril structure from patient FOR006[14]. The ordered core is decorated, in all fibrils, by relatively blurry density features of uncertain molecular origins. Only the FOR005 fibril contains a density that is resolved well enough to allow its assignment to a peripherally attached cross-β sheet structure[15].

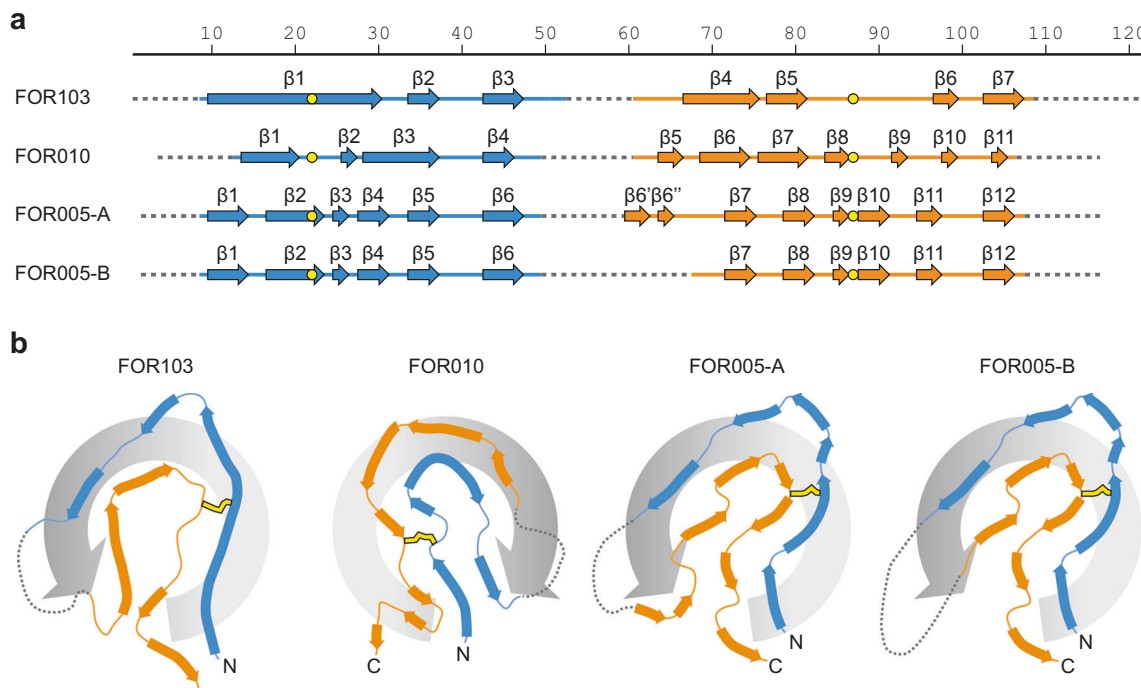

**Fig. 3 | Secondary structural composition of the fibrils. a** Location of the fibril cross-β structure in the sequence of the four fibril proteins. The numbering reflects the residue numbers of the respective precursor LCs after the removal of their signal sequences. Continuous lines refer to the ordered regions observed in the 3D map. Structurally disordered regions are shown as dotted lines. Cysteine residues participating in disulphide bonds are indicated in yellow. **b** Cross-sectional views of the four fibril proteins. The swirl of the amyloid key is indicated by the grey rotational arrow.

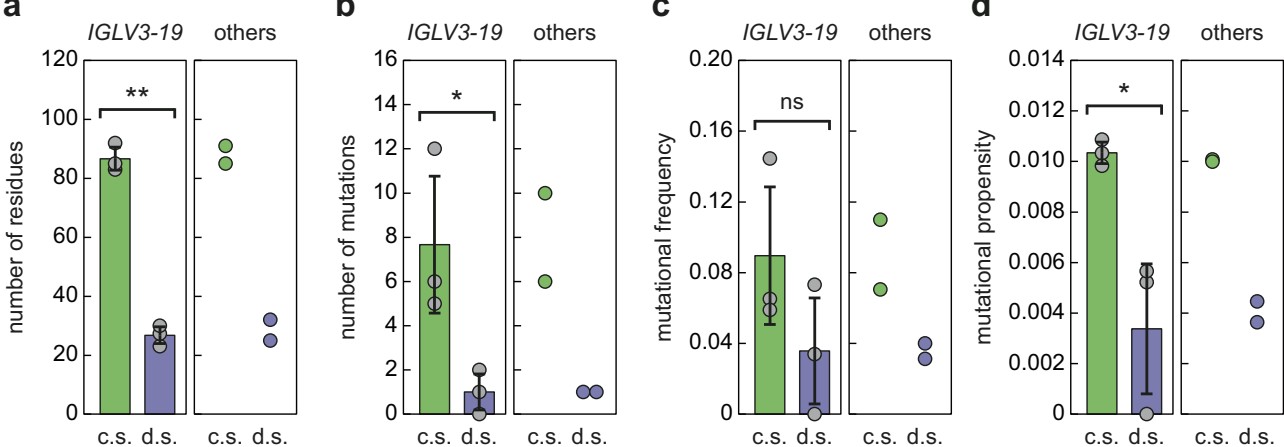

**Fig. 4 | Mutational propensities of the analysed fibril proteins. a** Length of the disordered segments (d.s.) and core segments (c.s.) of the presently analysed *IGLV3-19*-derived AL amyloid fibrils (left, *n* = 3, biological replicates) and of two other fibril structures (right, *n* = 2, biological replicates): FOR001 (*IGLV1-51*)[16] and FOR006 (*IGLV1-44*)[14]. **b** Number of the mutations in the two types of segments for the fibrils described in (**a**). **c** Mutational frequencies of these fibrils. **d** Mutational propensities of these fibrils. The colour coding is consistent in all panels. Grey data points: individual values of the *IGLV3-19*-derived AL amyloid fibrils. Bars, error bars: average values of these fibrils and standard deviation. Significant differences between the core and disordered segments are indicated by asterisks: p-value of 0.1 (*) and p-value of 0.01 (**). Non-significant (ns). Statistical analysis was performed using a two-sided *t* test. Source data are provided as a Source Data file.

## Role of the mutations for the fibril morphology

The fact that there are different fibrils structures associated with patients with a comparable phenotype (i.e., a prominent cardiac involvement) and *IGLV3-19*-derived AL amyloid fibrils (Fig. 2) implies that mutational changes contribute to determining the fibril structure. Support for this notion comes from the observation that the mutations occur twice as often in the stable fibril core than in the conformationally disordered segments (Fig. 4). The mutational propensity of the residues in the stable core of *IGLV3-19*-derived fibrils is 0.0103 ± 0.0004, while it is 0.0036 ± 0.0026 in the disordered segments. The difference between these values shows only a poor statistical significance (*p*-value 0.1), as the absence of mutations in the disordered segments of the FOR103 fibril protein leads to a very large standard deviation for the value in the disordered segments. However, a similar preference of the mutations for the stable core of the fibril is seen for two AL amyloid fibrils that are not derived from *IGLV3-19* segments (Fig. 4), indicating a preference of the mutations for the stable fibril core across all five patients.

Apart from this preference for the ordered segments we could not discern any clear trend in the location of the mutations within the fibril

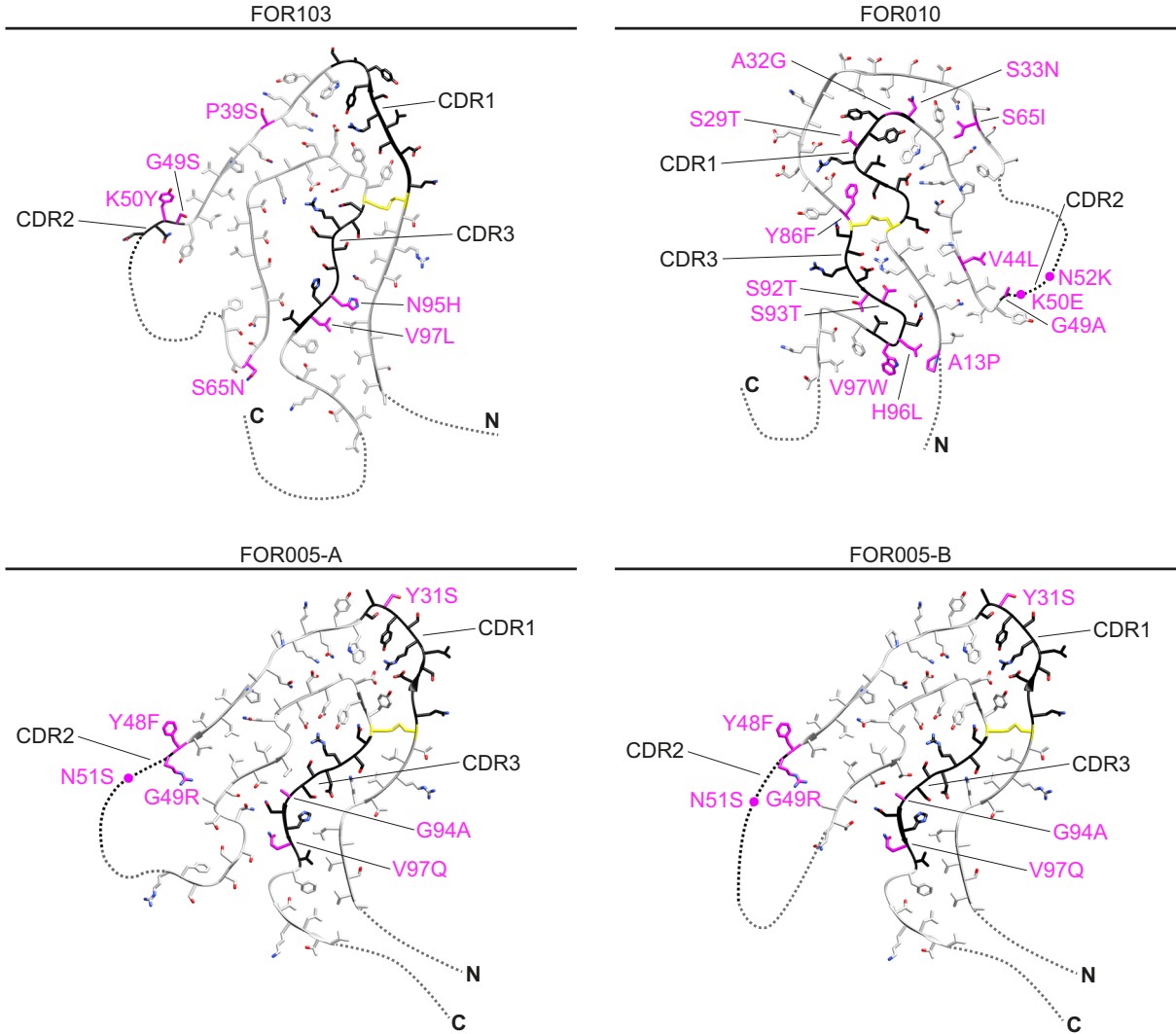

**Fig. 5 | Mutations of the fibril proteins.** Cross-sectional views of one molecular layer of the fibrils. Magenta: mutations compared to GL sequences of *IGLV3-19* and *IGLJ2* segments; yellow: cysteine residues involved in disulphide bond formation. Black: CDRs as described previously[16].

structures. Nor did the nature of the mutations suggest a simple and generally applicable explanation as to why the mutations might be structurally important (Fig. 5). There is only one mutational change in the analysed fibril structures that would intuitively be anticipated to favour the observed fibril protein fold. This mutation (H96L) affects a buried position of the FOR010 fibril, where it replaces a charged residue with a hydrophobic one. Yet, an almost reverse mutation (N95H) occurs in the FOR103 fibril protein, raising the question of whether the aforementioned H96L substitution does really contribute substantially to the observed fibril structure.

## Discussion

In the present study, we have explored as to whether or not the mutations in amyloidogenic LCs might affect the structures of the deposited fibrils. To that end, we compared the molecular structure of the amyloid fibrils from three patients with systemic AL amyloidosis (FOR103, FOR010 and FOR005). All patients suffered from LCs that were derived from an *IGLV3-19* gene segment. Using cryo-EM we find that the fibrils in the three patients share, first of all, a number of general similarities. Each fibril consists of a single fibril protein stack (Fig. 2). The fibril core is always formed from similar segments of the $V_L$ domain (Fig. 1). All fibril proteins contain an intramolecular disulphide bond with an amyloid key fold (Fig. 3). All fibril proteins show

conformations that differ starkly from the fold of natively folded LCs. Therefore, native LC conformations need to become unfolded as the proteins convert into fibrils. All fibril proteins show, compared to natively folded LCs, a flipped orientation of the β-strands at the disulphide bridge. This observation further corroborates the view that a rotational switch around the disulphide bond represents a key step discriminating native folding from the misfolding. In summary, our structures confirm previous conclusions that LC misfolding follows common mechanistic steps in different patients[14,16].

Besides these common features, however, we also find a number of important, patient-specific differences. These differences concern in particular the exact conformations of the fibril proteins and the overall twist of the fibrils (Figs. 2, 3), and they establish that the mutations underlying systemic AL amyloidosis may affect the fibril structure. Consistent with this view, we find that amyloidogenic mutations occur twice as often in the ordered parts of the fibril structure compared with the structurally disordered segments (Fig. 4d). We conclude that the role of LC mutations in systemic AL amyloidosis is not restricted to the previously described destabilising effect on the native LC conformation[3,9,22] and that they may also help to define the structure of the pathogenic agent. This conclusion does not imply that the mutations arose to allow the formation of a certain structure (as mutational changes occurred in evolution to allow the diversification of protein

structures). However, the dependence of the structure on the mutations may mean that only certain combinations of mutations allow the formation of a LC-derived fibril structure that is stable enough to become pathogenic in vivo. So far, however, it is unclear how the mutations in the investigated fibrils may lead to the observed structures (Fig. 5), and more detailed structural analyses will be required to reveal these effects.

Our current data relate to recently published cases of mutations in ex vivo amyloid fibrils. In the case of Aβ(1–42) peptide, it was found that a single-site substitution (E22G) suffices to change the fibril structure[23], while in the case of systemic ATTR amyloidosis, where the amyloid is derived from the protein transthyretin, fibrils seem to be much more tolerant to mutational changes, and consistent amyloid fibril structures were found in patients affected by different single-site mutations[24]. Transthyretin-derived fibril proteins are two- to three-fold larger than Aβ(1–42) peptide, indicating that a single-site change may be less effective in changing the fibril fold. LC-derived fibril proteins, by contrast, have about the size of the transthyretin-derived fibril proteins, but amyloidogenic LCs are typically affected by more than only a single-site mutational change - in our case there are six to fourteen mutations - and it is plausible that LCs can be more strongly affected by the mutations than transthyretin-derived fibrils.

Still, it is surprising that so few mutations suffice to alter the fibril structure, specifically if we compare the mutational effect in amyloid formation with the mutational effect in native protein folding reactions. Native protein folding reactions are much more tolerant to mutational changes, and changing only a few residues does not result, for a typical globular protein, in a substantially revised protein structure. Consistent with this, the native structures of amyloidogenic LCs share the same fundamental fold and there is no strong effect of the mutations on the globular protein conformation[17,25]. This tolerance of globular protein structures to mutational changes has been of general biological importance as it allowed the evolution of protein sequences without necessarily changing the protein's general structure and fold. In addition, it constitutes the scientific basis for any attempt to predict the structure of a globular protein based on its sequential homology to a known protein structure. The now documented strong effect of mutations on the structures of pathogenic amyloid fibrils highlights the general complexity of aggregation reactions and their fundamental differences from native folding reactions. As a result, it may be considerably more challenging to correctly predict the exact structures of pathogenic amyloid fibrils than the structure of globular proteins, where closely homologous sequences result in the formation of consistent tertiary structures.

## Methods
### Ethical statement
The patient's consent was obtained before the collection of the biopsies as approved by the Ethical Committee of the University of Heidelberg (S-123/2006). Fibril extraction and biochemical analysis were performed under valid permission from the Ethics Committee of Ulm University (203/18).

### Source of AL FOR103 fat tissue and AL FOR010 heart tissue
Fat tissue was collected from a female patient in her 80 s (FOR103) with AL amyloidosis and major heart involvement. Heart tissue was collected from a male patient in his 60 s (FOR010) with AL amyloidosis and advanced cardiac involvement. The tissue samples were stored at – 80 °C.

### Extraction of FOR103 amyloid fibrils from abdominal fat tissue
A 125 mg piece of abdominal fat tissue from patient FOR103 was placed in 250 μL of ice-cold tris(hydroxymehtyl)aminomethane (Tris) calcium buffer (20 mM Tris, 138 mM NaCl, 2 mM CaCl$_2$, 0.1 (w/v) % NaN$_3$, pH 8.0). The sample was inverted for 10 s and centrifuged for 5 min at 3100 × $g$ and 4 °C. The supernatant was removed with a syringe to

bypass the layer of fat that arose after centrifugation. The washing and centrifugation cycle was repeated four more times, always retaining the pellet and the superficial fat layer. The pellet was then resuspended in 0.5 mL of freshly prepared solution of 2 mg mL$^{-1}$ *Clostridium histolyticum* collagenase (Sigma) in Tris calcium buffer, containing ethylenediaminetetraacetic acid (EDTA)-free protease inhibitor (Roche), and incubated overnight at 37 °C. The incubation was followed by a centrifugation step of the sample at 3100 × $g$ for 30 min and the subsequent removal of the supernatant and fat layer. The resulting pellet was subjected to four washing and centrifugation steps with 250 μL Tris EDTA buffer (20 mM Tris, 140 mM NaCl, 10 mM EDTA, 0.1 % (w/v) NaN$_3$, pH 8.0) for 5 min at 3100 × $g$ and 4 °C. Finally, the resulting pellet was subjected to seven washing and centrifugation steps with 125 μL ice-cold water for 5 min at 3100 × $g$ and 4 °C. All supernatants were stored at 4 °C. The purification of FOR010 fibril protein is described in the Supplementary Information.

### Cryo-electron microscopy
A sample volume of 3.5 μL fibril solution was applied on glow discharged C-flat 1.2/1.3 400-mesh holey carbon-coated grids (Science Services). Plunge-freezing of the grids was performed with an EM GP2 Automatic Plunge Freezer (Leica) with an incubation time of 30 s at > 95 % humidity, back-side blotting using filter paper (Whatman) and plunging into liquid ethane. Afterwards, the grids were quality checked with a 200 kV JEM 2100 transmission electron microscope (JEOL), using a CMOS camera (TVIPS). Data sets for reconstruction were collected at 300 kV with a Titan Krios TEM (Thermo Fisher Scientific), using a K2 summit camera (Gatan, FOR103 fibrils) or a Falcon i4 camera (Thermo Fisher Scientific, FOR010 fibrils). The full data acquisition parameters are given in Supplementary Table 2.

### Helical reconstruction
The movie frames were gain corrected with IMOD[26] in the case of the FOR103 data set and with RELION 3.1.3[27] in case of the images of FOR010 fibrils. Both data sets were motion-corrected as well as dose-weighted with MotionCor2[28]. CTFFIND 4.1[29] was used to estimate the contrast transfer function (CTF) of the motion-corrected micrographs. The helical reconstruction was performed with RELION 3.1.3 for the FOR010 data set and with RELION 3.1.3 and RELION 5.0 for the FOR103 data set.

In the case of the FOR103 fibril, manually picked particles were extracted with a box size of 300 pixels. A reference-free two-dimensional (2D) classification was performed for 25 iterations, using a regularisation parameter of $T$ = 2. 2D classes with a well-resolved ~ 4.7 Å spacing along the fibril $z$-axis (SI Fig. 10a) were selected and an initial 3D map was reconstructed, using a featureless cylinder created with the relion_helix_toolbox as a reference. Based on the resulting 3D map, several rounds of 3D classification, particle selection and 3D refinement were performed (SI Fig. 10b). The final 3D map was masked with a soft edge mask and post-processed. The refinement was repeated with RELION 5.0, using the "blush regularisation" feature which increases the signal-to-noise ratio of the micrographs[30]. The resulting 3D map was polished with a Bayesian approach and subjected to CTF refinement. The final 3D map was masked with a soft edge mask and post-processed (SI Fig. 10c). The fibril was initially reconstructed with a left-hand twist and the correct handedness was imposed on the post-processed 3D map using the relion_image_handler.

In the case of the FOR010 fibril, particles were manually picked and extracted initially with a box size of 300 pixels. Reference-free 2D classification led to classes with well-resolved ~ 4.7 Å spacing along the fibril $z$-axis (SI Fig. 11a). A first 3D refinement was performed, using a featureless cylinder as a reference. Subsequent 3D classification, particle selection and 3D refinement steps were used to improve the 3D map. In the next step, particles were extracted again with a box size of 200 pixels and a 3D refinement was performed, using the

aforementioned 3D map as a reference. This refinement resulted in a 3D map, which was subjected to CTF refinement and Bayesian polishing, followed by another 3D classification with three classes and a regularisation parameter of $T = 4$ (SI Fig. 11b). The particles of the best 3D class were selected and a 3D refinement was performed. The resulting 3D map was improved with two consecutive rounds of CTF refinement and Bayesian polishing before it was masked with a soft edge mask and post-processed (SI Fig. 11c).

### Fitting of a molecular model to the 3D map

The 3D map of FOR103 was used to create one layer of the molecular model of the fibril in Coot[31]. The 3D map of FOR010 was used to automatically build one layer of the molecular model of the fibril, using the phenix implementation Autobuild. Modelling of both structures was performed in Coot on one molecular layer with phenix.real_space_refine[32]. Restraints imposed during modelling were non-crystallographic symmetry restraints, rotamer outliers, Ramachandran outliers and atomic geometry restraints. The validation of the molecular models after each round of modelling was performed with MolProbity[33], using rotamer outliers, Ramachandran outliers, atomic clashes and atomic geometry restraints. The cycle of modelling and validation was repeated until a satisfactory fit of the molecular model to the 3D map was achieved on one molecular layer. Afterwards, a six-layer structure was built with the Situs implementation pdbsymm[34]. Modelling and validation were repeated until a satisfactory fit of the six-layer molecular model to the 3D map was achieved. Finally, EMRinger scores[35] were calculated to estimate the accuracy of the assigned peptide backbone to the 3D map.

### Mutational propensity

The propensity of mutations in the AL fibrils was calculated by dividing the number of mutations in the ordered and disordered segments of the fibril proteins by the number of amino acids in these segments. In these calculations we used the predominant fibril proteins reported for patients FOR010 (residues 4-116, Fig. 1), as well as FOR001 (residues 1-118)[16], FOR005 (residues 2-116)[15], FOR006 (residues 3-118)[14], FOR103 (residues 2-116, Fig. 1). The resulting mutational frequencies were normalised to the total number of mutations in each fibril protein to yield the mutational propensities. For the two fibril structures from patient FOR005 average values are used.

### Statistical analysis

Analysis of statistical significance was performed with a two-sided $t$ test, assuming unequal variances.

### Determination of mutations in the fibril proteins

The mutations of the FOR103, FOR010 and FOR005 fibril proteins were obtained by comparison of their sequences with *IGLV3-19\*01*, *IGLJ2\*01* and *IGLC2* GL sequences from the IMGT database (https://www.imgt.org/). In the case of the FOR001 fibril, *IGLV1-51\*01*, *IGLJ2\*01* and *IGLC2* GL sequences were used. In the case of the FOR006 fibril, we used the GL sequences of *IGLV1-44\*01*, *IGLJ3\*02* and *IGLC3*.

### Reporting summary

Further information on research design is available in the Nature Portfolio Reporting Summary linked to this article.

## Data availability

The cryo-EM images were deposited in the Electron Microscopy Public Image Achieve with the accession codes EMPIAR-11801 (FOR103) and EMPIAR-11802 (FOR010). The reconstructed 3D maps were deposited in the Electron Microscopy Data Bank with the accession codes EMD-19818 (FOR103) and EMD-18881 (FOR010). The coordinate files of the fibril models were deposited in the Protein Data Bank with the accession codes 9EME (FOR103) and 8R47

(FOR010). The models of the previously published structures of FOR005 were deposited in the Protein Data Bank with the accession codes 6Z1O (FOR005-A) and 6Z1I (FOR005-B). The source data of Fig. 4 and Supplementary Figs. 1, 6 and 8 are available in the source data file. All unique biological materials are available from the corresponding author upon request. However, the amount of tissue available from patients FOR103 and FOR010 is limited. Source data are provided with this paper.

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

## Acknowledgements

This work was funded by a grant from the Deutsche Forschungsgemeinschaft (FA 456/23, FA 456/27 and 456/28 to M.F., HA 7138/3 to C.H., HE 8472/1-1 to U.H. and SCHO 1364/2-1 to S.O.S). The authors acknowledge technical support from Simon Fromm (European Molecular Biology Laboratory, Heidelberg) and Paul Walther (Ulm University). Cryo-EM data collection was funded by iNEXT (Horizon 2020, European Union).

## Author contributions

S.K., P.B.P., S.B., J.B., L.K., N.K. and S.W. performed experiments. S.K., P.B.P., S.B., J.B., N.K., C.H., M.S. and M.F. analysed the data. U.H. and S.S. provided material and reagents. M.F. designed research. S.K., P.B.P. and M.F. wrote the paper with the support of all other authors.

## Funding

## Competing interests

The authors declare no competing interests.
