## [Peer Review File · Nature Communications]

Reviewers' Comments:

Reviewer #1:

Remarks to the Author:

This paper describes two new cryo-EM structures of amyloids extracted from heart or fat tissue of two individuals with AL amyloidosis (FOR010 and FOR103). The FOR103 structure is done well; the FOR010 may benefit from some more work but is likely also correct. The observation that the two structures were solved from different tissues (heart and fat) raises the question whether the tissue itself may determine fibril structure, although different mutations of the protein forming the fibrils may also be the reason why these structures are different. The latter is also supported by comparison with 2 other structures. Overall, the work is done well and written up clearly, and I support publication, provided the points below are adequately addressed.

Line 79: what is "shape up"?

Line 150: "resembles the Greek key of globular protein structures" As Greek-key motif have also been used to describe alpha-synuclein amyloids, and these don't show much resemblance to the filaments described here, it may be better to remove the Greek key wording here.

Line 153: "the swirl of the key" This is an undefined concept.

FOR103: from fat tissue; FAR010 from heart tissue: could different structures form in different tissues? (This may be hard to address experimentally.)

Figure 4: all panels lack Y-axis labels (plus units). Perhaps therefore I don't understand what is plotted here. Where do these data come from?

SI Fig 6: The cross-sections of the FOR103 map suggest that densities are not well separated along the Z-direction for the entire core region, suggesting the map represents a local minimum of refinement. It may be worth to re-do these refinements and try and make it better.

SI Fig 8: "The lose up to the fight" What does that mean?

SI Table 2: resolution of FOR103 model is only 4.1A, while resolution of the map was 3.5A. This suggests that the map may represent a local minimum and atomic modelling may suffer from artefacts, as also suggested by SI Fig 6. As mentioned above, a bit more work may make this a better structure.

Model-map FSC curves are missing from the SI.

Reviewer #2:

Remarks to the Author:

The manuscript addresses an essential question for understanding the structural bases of the AL fibril formation, with implications for the pathogenesis of light chain-derived (AL) amyloidosis: how somatic mutations, in interplay with structural factors encoded in the germline sequence, define the structure of AL fibrils. To address this question, the authors determined the cryo-EM structure of three ex-vivo AL fibrils, (FOR010, 197 FOR103, and FOR005), derived from the same lambda-3 variable domain gene segment: IGLV3-19. Adding further relevance to this study is that the three AL fibrils were recovered from patients with cardiac AL amyloidosis.

The cryo-EM analysis showed that, despite the common genetic origin, the three lambda-3 LCs adopted amyloid fold with significant differences in the arrangement of the polypeptide chain. One intriguing observation is that the handedness of the fibril supertwist and the handedness of the

swirl of the amyloid key motif were not correlated. This observation highlights the complexity of the forces that determine the amyloid fibril structure. The authors concluded that the mutations of amyloidogenic light chains contribute to defining the fibril architecture and, hence, the structure of the pathogenic agent.

This study's novelty is based on the scientific problem addressed and the optimal experimental strategy implemented. The selection of appropriate samples (ex vivo AL fibrils encoded by the same VL gene segment) and the application of cryo-EM analysis, currently the most powerful procedure to study amyloid aggregates, allowed the authors to generate data of high scientific relevance. The effect of somatic mutations on the biophysical properties of light chains has been widely studied, and numerous studies have been published on this topic. But it has been only recently that we began to understand how these changes influence the structure of AL fibrils. Addressing this question requires a totally different experimental approach than that used in studies with the native light chain, due to the marked structural differences between the native and the amyloid state. That's why this manuscript represents a significant methodological contribution to addressing the role of somatic mutations in AL fibril structure.

Furthermore, this study provides clear evidence of the ability of somatic mutations to define the amyloid conformation of light chains, which deepens our understanding of the determinants of the structural heterogeneity of AL fibrils. Given that evidence indicates that AL fibrils play a key role in the pathogenic mechanisms causing organ dysfunction, the knowledge generated in this study also has implications for understanding the basis of the clinical heterogeneity of AL amyloidosis.

The manuscript is written clearly and coherently, with good use of language. The figures have an adequate design and transmit structural information effectively. The references cited are adequate in number and relevance. Including an additional reference in the introduction, which is optional, has been suggested.

In conclusion, this is a high-quality study, reported in a well-written manuscript. I have found some minor issues that are listed below:

1) I agree with the authors that the most likely cause of the structural differences between the three lambda-3 AL fibrils are the somatic mutations since most of the changes are in the VL segments that form the ordered fibril core. But this is what can be expected, given that the CDR1 and CDR3 are part of these segments. In almost all AL fibrils studied, the N- and C-terminus and part of the loop that forms the VH-VL interface, called the 40-60 loop, are in the conformationally disordered regions. These segments, excepting the short CDR2 in the 40-60 loop, fold into the framework regions of the VL domain, so they tend to contain fewer mutations than the CDRs. Considering this, the mutation propensity analysis will necessarily identify the regions that form the ordered fibril core as the preferred target of mutations. My concern is that the explanation of the mutation propensity analysis between lines 174 and 181 may suggest to some readers that the preferential localization of the mutations in the VL segments that form the ordered core of the AL fibrils responds to structural requirements of these aggregates. What seems more likely is that the conformational constraints imposed by the CL and the presence of up to 4 conserved prolines in the 40-60 loop make the C-terminal and the 40-60 loop less suitable than the rest of the VL (despite the greater number of somatic mutations) to form the ordered core of the AL fibril. I would like to know the author's opinion on this.

2) In line 30 and again in line 41, the authors state "Systemic AL amyloidosis is one of the most abundant forms of systemic amyloidosis." Abundant is an adjective that describes something present in large quantities and is not normally used to describe the disease's prevalence. AL amyloidosis is a rare disease since its incidence rate and prevalence has been calculated at 1.2 per 100,000 person-years (1) and 40.5 cases per million(2). Using "abundant" may suggest a higher prevalence of this disease to readers unfamiliar with this disorder. A more accurate wording may be "Systemic AL amyloidosis is one of the most frequently diagnosed forms of systemic amyloidosis."

3) For those readers who are not familiar with monoclonal gammopathies, the statement "The disease arises from the misfolding of antibody light chains (LC)," lines 41-42, may suggest that the misfolding of polyclonal light chains causes AL amyloidosis. Although in line 50 the authors say, "Almost every patient is affected by a sequentially different LC," the monoclonal origin of the AL amyloidogenic may not be clear to all readers. It is important to highlight that AL amyloidosis

arises from misfolding and aggregation of a monoclonal light chain.

4) The reference cited at the end of line 46 could be accompanied by a recently published review that specifically addresses the role of mechanisms for antibody repertoire diversification in monoclonal light chain deposition disorders (3).

([1.] Kyle, R. A., Larson, D. R., Kurtin, P. J., Kumar, S., Cerhan, J. R., Therneau, T. M., Rajkumar, S. V., Vachon, C. M., and Dispenzieri, A. (2019) Incidence of AL Amyloidosis in Olmsted County, Minnesota, 1990 through 2015, *Mayo Clin Proc* 94, 465-471.

([2.] Quock, T. P., Yan, T., Chang, E., Guthrie, S., and Broder, M. S. (2018) Epidemiology of AL amyloidosis: a real-world study using US claims data, *Blood Adv* 2, 1046-1053.

([3.] Del Pozo-Yauner, L., Herrera, G. A., Perez Carreon, J. I., Turbat-Herrera, E. A., Rodriguez-Alvarez, F. J., and Ruiz Zamora, R. A. (2023) Role of the mechanisms for antibody repertoire diversification in monoclonal light chain deposition disorders: when a friend becomes foe, *Front Immunol* 14, 1203425.

Reviewer #3:

Remarks to the Author:

In this study, Karimi-Farsijani and co-workers aimed to scrutinize the impact of mutational changes in immunoglobulin light chains on fibril structures in patients with cardiac AL amyloidosis. This is an important topic as the hypervariability of antibody LCs introduces unique mutations in each patient, ultimately leading to distinct pathological manifestations. The Fändrich group utilized cryo-electron microscopy to compare fibril structures, focusing on a common IGLV3-19 gene segment. The study encompassed two previously published fibrils from a single patient from the same group, alongside two new fibrils from distinct patients. Despite significant overlap in the involved β -sheet regions, all fibrils exhibited distinct architectures, varying in conformations and overall twist.

The authors convincingly illustrate that beyond influencing misfolding propensity or stability in the native state, mutations also impact the folding of the polypeptide chain in the fibrillar state. This is reinforced by the observation that mutations tend to localize in structured regions of the fibrils.

The mechanisms through which mutations contribute to specific structures remain unclear and challenging to decipher. However, it would be valuable to investigate whether mutations alter the aggregation propensities of the corresponding sequences using algorithms like Aggrescan, CamSol, or others while exploring and discussing whether aggregation-prone sequences map or not to the different structured beta-sheets in the fibrils.

Additionally, understanding how a given fibril structure correlates with a specific clinical phenotype remains elusive and, again, difficult to decipher. Exploring the physicochemical properties of the different fibrils, especially those of the lateral surfaces likely interacting with other macromolecules in cells/tissues, might provide interesting insights. Calculating local and global hydrophobicity/polarity of these lateral regions, as well as evaluating these values for the upper/lower faces of the fibrils where elongation occurs, and seeing whether the fibrils are similar/different in these properties would be very interesting.

Another essential factor influencing fibril physiological behavior might be their relative thermodynamic stability. Theoretical approximations of these values using programs like Rosetta or Fold-X would be of interest since, as a general trend, pathogenic fibrils tend to be more stable and hydrophobic than functional ones.

Overall, this work is an interesting contribution to our knowledge of amyloid fibril structure. Still, I think that the proposed analysis might allow to go a bit deeper into the mechanistic aspects of the

sequence/structure relationship.

Revision notes

Reviewer #1 (Remarks to the Author):

This paper describes two new cryo-EM structures of amyloids extracted from heart or fat tissue of two individuals with AL amyloidosis (FOR010 and FOR103). The FOR103 structure is done well; the FOR010 may benefit from some more work but is likely also correct. The observation that the two structures were solved from different tissues (heart and fat) raises the question whether the tissue itself may determine fibril structure, although different mutations of the protein forming the fibrils may also be the reason why these structures are different. The latter is also supported by comparison with 2 other structures. Overall, the work is done well and written up clearly, and I support publication, provided the points below are adequately addressed.

Line 79: what is “shape up”?

Response: We thank the referee for thoroughly analysing the data and for providing these helpful comments which allowed us to further improve the manuscript. The phrase that the mutations “shape up” the structure of the pathogenic agent was supposed to mean that the mutations are a defining factor in determining the structure of the amyloid fibril and lead to different, patient specific fibril protein folds. We now changed the wording to “Hence, they play a crucial role in defining the structure”.

Line 150: “resembles the Greek key of globular protein structures” As Greek-key motif have also been used to describe alpha-synuclein amyloids, and these don’t show much resemblance to the filaments described here, it may be better to remove the Greek key wording here.

Response: Thank you for this suggestion. We rephrased the text to “The amyloid key fold resembles the Greek key motif of globular protein structures. However, the amyloid key does not form intramolecular β -sheets, but is instead stabilised by intermolecular hydrogen bonds (β -sheets) along the fibril z-axis and side chain-side chain interactions of the β -strands.” and added a reference (Sharma et al., 2024, PMID: 38212334) to avoid confusion between “amyloid key” and “Greek key” motifs.

Line 153: “the swirl of the key” This is an undefined concept.

Response: The term “swirl” was previously used in the protein structure field in the context of beta-barrels and jellyroll structures, but not so much in the context of amyloid fibrils. We here chose the term “swirl” to avoid confusion between the handedness of the fibril and of the amyloid key fold. We now define the term “swirl” in the text by adding the following explanation “The swirl is defined by the position of the central arc of the amyloid key relative to the N- and C-terminal ends of the polypeptide chain”.

FOR103: from fat tissue; FAR010 from heart tissue: could different structures form in different tissues? (This may be hard to address experimentally.)

Response: Previous analyses of AL amyloid fibrils from different tissues of the same patient does not support this conjecture. We found the same fibril morphology in heart tissue, abdominal fat and heart fat (Annamalai et al., 2017, PMID: 28544119). Similarly, the Ricagno group reports the same fibril structures and protein folds in the kidney and heart of an AL patient (Puri et al., 2023, PMID: 37516426). In case of patient FOR103 we were only able to obtain samples of abdominal fat but no heart tissue.

Figure 4: all panels lack Y-axis labels (plus units). Perhaps therefore I don’t understand what is plotted here. Where do these data come from?

Response: As requested, we have added labels to figure 4. The data come from the analysis of the mutations of the fibril proteins compared to the GL sequence. The mutational frequency normalised the mutations in the ordered and disordered regions of the protein by the length of two regions. The mutational propensity is the mutational frequency normalised by the total number of mutations in the two regions, respectively. For further information, please refer to the methods section.

SI Fig 6: The cross-sections of the FOR103 map suggest that densities are not well separated along the Z-direction for the entire core region, suggesting the map represents a local minimum of refinement. It may be worth to re-do these refinements and try and make it better.

Response: Although this request caused extra work for us, it was actually worthwhile and we thank the referee for this suggestion. We repeated the refinements and were able to perform Bayesian polishing and CTF refinement with RELION 5.0 using the “blush regularisation” feature during 3D refinement. This has increased the resolution to 2.92 Å, and significantly improved the separation of the fibril z-axis as well as the cross-section. We added a paragraph to the methods section, describing the reconstruction of the FOR103 structure with RELION 5.0 and the new reconstruction parameters were added to SI Table 2. SI Figure 6, 7 and 9 were changed to include the new 3D map and FSC curves. The EMD entry has changed accordingly to 19818.

SI Fig 8: “The lose up to the fight” What does that mean?

Response: We thank the referee for spotting this corrupt sentence, which has been corrected to “The close up to the right...”.

SI Table 2: resolution of FOR103 model is only 4.1Å, while resolution of the map was 3.5Å. This suggests that the map may represent a local minimum and atomic modelling may suffer from artefacts, as also suggested by SI Fig 6. As mentioned above, a bit more work may make this a better structure.

Response: As suggested by the reviewer, we obtained a new reconstruction and redid the modelling. The model resolution is now 3.1 Å. The new modelling parameters were added to SI Table 3. Figure 2, 3 and 5 as well as SI Figure 7 and 9 were changed to include the new model and changed beta-sheet assignment. The PDB entry has changed accordingly to 9EME.

Model-map FSC curves are missing from the SI.

Response: The model-map FSC curves were added to the SI as SI Figure 8.

Reviewer #2 (Remarks to the Author):

The manuscript addresses an essential question for understanding the structural bases of the AL fibril formation, with implications for the pathogenesis of light chain-derived (AL) amyloidosis: how somatic mutations, in interplay with structural factors encoded in the germline sequence, define the structure of AL fibrils. To address this question, the authors determined the cryo-EM structure of three ex-vivo AL fibrils, (FOR010, 197 FOR103, and FOR005), derived from the same lambda-3 variable domain gene segment: IGLV3-19. Adding further relevance to this study is that the three AL fibrils were recovered from patients with cardiac AL amyloidosis. The cryo-EM analysis showed that, despite the common genetic origin, the three lambda-3 LCs adopted amyloid fold with significant differences in the arrangement of the polypeptide chain. One intriguing observation is that the handedness of the fibril supertwist and the handedness of the swirl of the amyloid key motif were not correlated. This observation highlights the complexity of the forces that determine the amyloid fibril structure. The authors concluded that the mutations of amyloidogenic light chains contribute to defining the fibril architecture and, hence, the structure of the pathogenic agent. This study's novelty is based on the scientific problem addressed and the optimal experimental strategy implemented. The selection of appropriate samples (ex vivo AL fibrils encoded by the same VL gene segment) and the application of cryo-EM analysis, currently the most powerful procedure to study amyloid aggregates, allowed the authors to generate data of high scientific relevance. The effect of somatic mutations on the biophysical properties of light chains has been widely studied, and numerous studies have been published on this topic. But it has been only recently that we began to understand how these changes influence the structure of AL fibrils. Addressing this question requires a totally different experimental approach than that used in studies with the native light chain, due to the marked structural differences between the native and the amyloid state. That's why this manuscript represents a significant methodological contribution to addressing the role of somatic mutations in AL fibril structure. Furthermore, this study provides clear evidence of the ability of somatic mutations to define the amyloid conformation of light chains, which deepens our understanding of the determinants of the structural heterogeneity of AL fibrils. Given that evidence indicates that AL fibrils play a key role in the pathogenic mechanisms causing organ dysfunction, the knowledge generated in this study also has implications for understanding the basis of the clinical heterogeneity of AL amyloidosis.

The manuscript is written clearly and coherently, with good use of language. The figures have an adequate design and transmit structural information effectively. The references cited are adequate in number and relevance. Including an additional reference in the introduction, which is optional, has been suggested. In conclusion, this is a high-quality study, reported in a well-written manuscript. I have found some minor issues that are listed below:

- 1) I agree with the authors that the most likely cause of the structural differences between the three lambda-3 AL fibrils are the somatic mutations since most of the changes are in the VL segments that form the ordered fibril core. But this is what can be expected, given that the CDR1 and CDR3 are part of these segments. In almost all AL fibrils studied, the N- and C-terminus and part of the loop that forms the VH-VL interface, called the 40-60 loop, are in the conformationally disordered regions. These segments, excepting the short CDR2 in the 40-60 loop, fold into the framework regions of the VL domain, so they tend to contain fewer mutations than the CDRs. Considering this, the mutation propensity analysis will necessarily identify the regions that form the ordered fibril core as the preferred target of mutations. My concern is that the explanation of the mutation propensity analysis between lines 174 and 181 may suggest to some readers that the preferential localization of the mutations in the VL segments that form the ordered core of the AL fibrils responds to structural requirements of these aggregates. What

seems more likely is that the conformational constraints imposed by the CL and the presence of up to 4 conserved prolines in the 40-60 loop make the C-terminal and the 40-60 loop less suitable than the rest of the VL (despite the greater number of somatic mutations) to form the ordered core of the AL fibril. I would like to know the author's opinion on this.

Response: We thank this referee for thoroughly analysing our data and for commenting so positively on our study. We agree that our data do not mean that there has been any biological mechanism that has optimized certain light chain sequences to allow the formation of a specific structure. Instead, we think that certain mutations happen to allow the formation of fibril morphologies that are stable enough to eventually become pathogenic. We now address this topic in the revised discussion “This conclusion does not imply that the mutations arose to allow the formation of a certain structure (as mutational changes occurred in evolution to allow the diversification of protein structures). However, it is possible that the dependence of the structure on the mutations means that only certain combinations of mutations allow the formation of a LC-derived fibril structure that is stable enough to become pathogenic in vivo.”.

2) In line 30 and again in line 41, the authors state "Systemic AL amyloidosis is one of the most abundant forms of systemic amyloidosis." Abundant is an adjective that describes something present in large quantities and is not normally used to describe the disease's prevalence. AL amyloidosis is a rare disease since its incidence rate and prevalence has been calculated at 1.2 per 100,000 person-years (1) and 40.5 cases per million(2). Using “abundant” may suggest a higher prevalence of this disease to readers unfamiliar with this disorder. A more accurate wording may be "Systemic AL amyloidosis is one of the most frequently diagnosed forms of systemic amyloidosis."

Response: Thank you for this comment. We changed lines 30 and 41 as suggested.

3) For those readers who are not familiar with monoclonal gammopathies, the statement “The disease arises from the misfolding of antibody light chains (LC),” lines 41-42, may suggest that the misfolding of polyclonal light chains causes AL amyloidosis. Although in line 50 the authors say, "Almost every patient is affected by a sequentially different LC," the monoclonal origin of the AL amyloidogenic may not be clear to all readers. It is important to highlight that AL amyloidosis arises from misfolding and aggregation of a monoclonal light chain.

Response: We thank the referee for pointing out this issue. We have adapted the term monoclonal LCs as the cause of AL amyloidosis in the introduction to avoid confusion with polyclonal LCs.

4) The reference cited at the end of line 46 could be accompanied by a recently published review that specifically addresses the role of mechanisms for antibody repertoire diversification in monoclonal light chain deposition disorders (3).

(1.) Kyle, R. A., Larson, D. R., Kurtin, P. J., Kumar, S., Cerhan, J. R., Therneau, T. M., Rajkumar, S. V., Vachon, C. M., and Dispenzieri, A. (2019) Incidence of AL Amyloidosis in Olmsted County, Minnesota, 1990 through 2015, *Mayo Clin Proc* 94, 465-471.

(2.) Quock, T. P., Yan, T., Chang, E., Guthrie, S., and Broder, M. S. (2018) Epidemiology of AL amyloidosis: a real-world study using US claims data, *Blood Adv* 2, 1046-1053.

(3.) Del Pozo-Yauner, L., Herrera, G. A., Perez Carreon, J. I., Turbat-Herrera, E. A., Rodriguez-Alvarez, F. J., and Ruiz Zamora, R. A. (2023) Role of the mechanisms for antibody repertoire diversification in monoclonal light chain deposition disorders: when a friend becomes foe, *Front Immunol* 14, 1203425.

Response: We thank the referee for this valuable suggestion and have now cited the above-mentioned review (Pozo-Yauner, 2023, PMID: 37520549), which addresses the hypervariability of LCs in the context of AL amyloidosis.

Reviewer #3 (Remarks to the Author):

In this study, Karimi-Farsijani and co-workers aimed to scrutinize the impact of mutational changes in immunoglobulin light chains on fibril structures in patients with cardiac AL amyloidosis. This is an important topic as the hypervariability of antibody LCs introduces unique mutations in each patient, ultimately leading to distinct pathological manifestations. The Fändrich group utilized cryo-electron microscopy to compare fibril structures, focusing on a common IGLV3-19 gene segment. The study encompassed two previously published fibrils from a single patient from the same group, alongside two new fibrils from distinct patients. Despite significant overlap in the involved β -sheet regions, all fibrils exhibited distinct architectures, varying in conformations and overall twist. The authors convincingly illustrate that beyond influencing misfolding propensity or stability in the native state, mutations also impact the folding of the polypeptide chain in the fibrillar state. This is reinforced by the observation that mutations tend to localize in structured regions of the fibrils.

The mechanisms through which mutations contribute to specific structures remain unclear and challenging to decipher. However, it would be valuable to investigate whether mutations alter the aggregation propensities of the corresponding sequences using algorithms like Aggrescan, CamSol, or others while exploring and discussing whether aggregation-prone sequences map or not to the different structured beta-sheets in the fibrils.

Response: We thank the referee for thoroughly analysing our data and providing helpful comments and suggestions. As proposed, we evaluated the sequences with different methods. We here show the data using Aggrescan, Pasta 2.0 and Waltz (Figure R1) but other programs (Tango 2.2, Amylpred and Foldamyloid) produced a similar outcome. Overall there is only a poor, if any correlation, of the predictions with the observed structures. The beta-sheets structure is only predicted with low confidence (typically less than 50 % correctly predicted residues). A random assignment would get 50 % of the residues correct. Yet, the predictions are probably better than pure chance and indeed perform better in the non-beta segments. Nevertheless, there is no convincing correlation of the aggregation-prone segments with the experimental structure, suggesting that there are other factors than just the general aggregation propensity that determine a pathologically relevant fibril structure. Since we have reported this observation several times in recent publications (Rademaker et al., 2021, PMID: 33558536; Rademaker et al., 2021, PMID: 34741031; Sharma et al., 2023, PMID: 37481159; Sharma et al., 2024, PMID: 38212334), we see no real benefit to report it again in the current study. Our concerns on this issue are also generally accessible through this revision letter.

Figure R1

Figure R1. Location of the fibril β -sheets and predicted aggregation prone segments by different programs. Dots indicate that the respective program assigns the residues as aggregation prone.

Figure R2

Figure R2. Comparison of the aggregation prone segments with the experimental structure. True positive: an aggregation prone residue participates in the formation of a β -sheet; false negative: a residue is in β -sheet conformation but not recognized by a program as aggregation prone; false positive: a residue is assigned as aggregation prone but is not part of a β -sheet; true negative: a residue is neither recognised as aggregation prone and nor involved in β -sheet formation.

Additionally, understanding how a given fibril structure correlates with a specific clinical phenotype remains elusive and, again, difficult to decipher. Exploring the physicochemical properties of the different fibrils, especially those of the lateral surfaces likely interacting with other macromolecules in cells/tissues, might provide interesting insights. Calculating local and

global hydrophobicity/polarity of these lateral regions, as well as evaluating these values for the upper/lower faces of the fibrils where elongation occurs, and seeing whether the fibrils are similar/different in these properties would be very interesting.

Response: We have evaluated the fibril structures with respect to the distribution of hydrophobic and polar chemical groups (Figure R3). As expected for different sequences and fibril structures, there are differences in the distribution of the chemical groups. Yet, it is difficult for us to generate any robust conclusion from these variations, as the existence of variations is not as such surprising. Perhaps it is possible to extract some useful information from such kind of analyses if carried out with a large number of fibrils and by comparison of systematically distinct groups of fibrils, such as *ex vivo* and *in vitro* formed fibrils. However, this type of analysis is beyond the scope of the present study.

Figure R3

Figure R3. Comparison of the chemical properties of the fibril proteins. Purple: polar surfaces; white: neutral surfaces; yellow: hydrophobic surfaces.

Another essential factor influencing fibril physiological behavior might be their relative thermodynamic stability. Theoretical approximations of these values using programs like Rosetta or Fold-X would be of interest since, as a general trend, pathogenic fibrils tend to be more stable and hydrophobic than functional ones.

Response: We have carried out theoretical stability approximations. It should be borne in mind, however, that each method has certain downsides or may not consider all contributions that are important for the true thermodynamic stability of a fibril. For example, ePISA calculations look only at intermolecular interactions between two molecules but not within a molecule. Furthermore, what about entropy and water effects? The other general problem with the existing theoretic approximations is that there have not been very many experimental measurements on the thermodynamic stability of fibrils - and in particular when relating them to single fibril morphologies. Hence, there is no good confidence on the relationship of theoretic approximations and real fibril stabilities. In case of the presently analysed fibrils the values calculated by different methods differ in magnitude and in their relative order (Figure R4). Of course, we could take these numbers face value and interpret them. But would we believe conclusions derived from this? Probably not.

Figure R4

Figure R4. Stability calculations of the fibril proteins with different programs.

Overall, this work is an interesting contribution to our knowledge of amyloid fibril structure. Still, I think that the proposed analysis might allow to go a bit deeper into the mechanistic aspects of the sequence/structure relationship.

Response: Again, we thank the referee for the supportive comments. We also agree that the points raised by this referee could potentially reveal interesting relationships, when carried out for appropriate large and well-defined cohorts of fibril structures. However, we report here two new fibril structures, along with two previously published ones, that all originated from similar sources (hearts of AL patients, who exhibited similar clinical characteristics).

Reviewers' Comments:

Reviewer #1:

Remarks to the Author:

This is OK now. It is satisfying that the authors were able to improve the FOR010 structure. They mention Blush regularisation played an important role. It would be good if they could cite the corresponding preprint: <https://www.biorxiv.org/content/10.1101/2023.10.23.563586v1>

(This preprint should soon come out in Nature Methods, so perhaps the preprint citation could be changed to the Nature Methods one, once it comes out.)

Reviewer #2:

Remarks to the Author:

The authors have thoroughly addressed all the suggestions and concerns I previously raised. The revisions have significantly enhanced the manuscript's clarity, facilitating a better understanding of the key findings from this compelling structural study. The data presented are of high quality, and the manuscript is well-prepared, ensuring that its contribution to the understanding of the structural bases of amyloid aggregation of light chains will be substantial and valuable.

Reviewer #3:

Remarks to the Author:

The authors performed all the requested calculations and analyses and provided solid responses to my questions and concerns.

Revision notes

Reviewer #1 (Remarks to the Author):

This is OK now. It is satisfying that the authors were able to improve the FOR010 structure. They mention Blush regularisation played an important role. It would be good if they could cite the corresponding preprint: <https://www.biorxiv.org/content/10.1101/2023.10.23.563586v1>

(This preprint should soon come out in Nature Methods, so perhaps the preprint citation could be changed to the Nature Methods one, once it comes out.)

Response: We now cite the pre-print when describing the reconstruction of the FOR103 structure with the Blush regularisation feature of RELION 5.0.

Reviewer #2 (Remarks to the Author):

The authors have thoroughly addressed all the suggestions and concerns I previously raised. The revisions have significantly enhanced the manuscript's clarity, facilitating a better understanding of the key findings from this compelling structural study. The data presented are of high quality, and the manuscript is well-prepared, ensuring that its contribution to the understanding of the structural bases of amyloid aggregation of light chains will be substantial and valuable.

Reviewer #3 (Remarks to the Author):

The authors performed all the requested calculations and analyses and provided solid responses to my questions and concerns.

Response: We thank all the referees for their thorough analysis of our manuscript and for providing helpful and constructive comments and suggestions.